# ReFoRM: Reliable Per-Base Error Prediction under Distribution Shifts in DNA Storage

## Abstract

Reliable prediction under distribution shift remains a core challenge in machine learning. In DNA storage pipelines, base-level errors are pervasive: synthesis introduces insertions and deletions, sequencing produces systematic substitutions, and amplification reshapes error distributions under stress conditions such as aging and PCR. Existing approaches often rely on scaling large sequence models or concatenating handcrafted descriptors, but both strategies suffer from redundancy, instability, and poor calibration. To address these issues, we propose ReFoRM, a lightweight framework for reliable prediction under distribution shift. Specifically, it consists of three components: (i) *feature refinement*, which selects a compact and informative subset from an over-complete feature pool; (ii) *cross-attentive fusion*, which integrates refined descriptors with embeddings in a stable and balanced manner; and (iii) *condition-aware calibration*, which adjusts predictive confidence under distribution shifts. We evaluate ReFoRM on DNA storage error prediction, where descriptors such as GC content, homopolymer length, and structural accessibility provide a natural testbed with perturbations. Across digital twin and simulated datasets, ReFoRM achieves **PR–AUC** 0.9278 (in); 0.9291 (LOCO) and **ECE** 0.0944 (in); 0.0968 (LOCO), demonstrating strong extensibility and reliability under distribution shifts.

## 1 Introduction

Reliable prediction under distribution shift is a central challenge in machine learning. Neural networks are often miscalibrated, generating overconfident probabilities even when accuracy declines (Guo et al., 2017). This issue becomes more severe under perturbations such as domain shift, where calibration quality degrades despite strong in-distribution performance (Ovadia et al., 2019). Such behavior not only reduces the trustworthiness of model predictions but also limits their applicability in scenarios where uncertainty estimates are critical. Consequently, robustness and calibration under shifting conditions are essential for deploying models in risk-sensitive applications, including domains such as healthcare, finance, and the natural sciences.

Synthetic biology provides a demanding testbed for examining these challenges. It has rapidly developed into a multidisciplinary field that spans DNA data storage, genetic circuit design, and large-scale oligonucleotide synthesis (Cao et al., 2022; Rai et al., 2024; Senan et al., 2024). These pipelines enable long-term information preservation, programmable cellular control, and scalable biomanufacturing (Zhang et al., 2024a; Teng et al., 2024; Sarkar et al., 2025). Despite major progress in throughput and cost reduction, reliability remains the key barrier that prevents laboratory prototypes from becoming robust engineering platforms (Lin et al., 2022; Marinelli et al., 2024). In particular, per-base errors introduced at different stages can accumulate and propagate, which undermines the fidelity of synthesized constructs and encoded information (Yeom et al., 2023).

The error landscape in synthetic biology is both complex and context dependent. Synthesis introduces insertion and deletion events because chemical reactions are not perfectly accurate. Sequencing platforms impose systematic substitution biases depending on the technology, and amplification procedures such as PCR reshape error distributions by preferentially amplifying certain sequence variants (Chen et al., 2020; Shi et al., 2021; Gimpel et al., 2023). These errors are not stationary: their frequency and type shift under stress conditions such as long-term aging or high-cycle PCR (Meiser et al., 2022). Certain motifs (e.g., homopolymer runs and GC-rich windows) and structural features

(e.g., secondary folding and accessibility) are disproportionately affected, leading to heterogeneous vulnerabilities across the sequence (Dwarshuis et al., 2024). As a result, global error-rate summaries are insufficient. Practitioners require *per-base risk prediction* to identify unstable regions and to proactively design robust encodings.

Existing approaches to reliability enhancement face critical limitations (Weindel et al., 2025). Coding-theoretic solutions emphasize redundancy and error correction, but they overlook intrinsic sequence biases that drive error formation (Welzel et al., 2023; Ding et al., 2024). Machine-learning methods have begun to model local sequence patterns, but they often generalize poorly when exposed to distribution shifts across experimental conditions (Angermueller et al., 2019; Zhang et al., 2023; Rafi et al., 2024). Their predictive probabilities are also frequently *miscalibrated*, which reduces their value for risk-aware decisions such as selective redundancy allocation or condition-specific reencoding (Fenlon et al., 2018; Arrieta-Ibarra et al., 2022; Tom et al., 2023). In addition, current models rarely integrate domain knowledge from structural and biophysical features, leaving a gap between predictive accuracy and actionable deployment. These limitations motivate the development of a framework that unifies refined feature representations, robust fusion mechanisms, and condition-aware calibration to enable trustworthy per-base error predictions (Zhang et al., 2024b; Kabir et al., 2024; Wang et al., 2025).

This paper introduces REFORM (see Figure 1), which is structured around three main components: **Re**fined features, **F**usion **of R**epresentations, and **M**odel calibration under condition shifts. Specifically: *(i) Feature refinement.* Starting from a pool of 51 sequence- and position-derived descriptors (e.g., k-mer context, homopolymer length, GC windows, and accessibility or thermodynamic surrogates), REFORM selects a compact 9-dimensional subset using SHAP attribution and PR–AUC drop analyses. Normalization schemes are applied to preserve cross-feature comparability (Lundberg & Lee, 2017). *(ii) Cross-attentive fusion.* The refined descriptors are combined with contextual token embeddings through a cross-attentive, convex vector-gated fusion module. This module produces

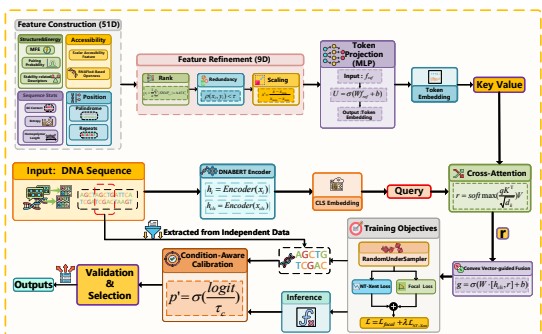

Figure 1: Overview of REFORM. Refined descriptors are embedded and fused with contextual representations via cross-attention and convex vector-gating. Condition-aware calibration is applied post hoc to improve reliability under perturbations.

bounded and stable representations, mitigating feature dominance and scale mismatches (Arevalo et al., 2017; Anil et al., 2019; Xu et al., 2021). *(iii) Condition-aware calibration.* To address distribution shifts across stress conditions, REFORM applies per-condition temperature scaling. Each condition is assigned a learnable temperature that monotonically maps logits to probabilities. This preserves rank-based discrimination (e.g., ROC–AUC invariance under monotone transforms) while improving calibration metrics such as ECE and Brier score (Guo et al., 2017; Ovadia et al., 2019).

We evaluate REFORM in the context of DNA storage pipelines, where per-base risk estimates are particularly important. Such predictions enable practical strategies including discarding or rewriting unstable substrings, allocating redundancy, and auditing constructs under aging and PCR protocols. In this way, REFORM demonstrates how the integration of descriptor refinement, gated fusion, and calibration can deliver reliable predictions that support downstream applications.

## 2 RELATED WORK

Errors in biological data pipelines have long been recognized, paralleling robustness concerns in other areas of machine learning. Large-scale DNA synthesis and assembly are prone to errors from incomplete nucleotide coupling and byproducts that introduce substitutions, insertions, or deletions (Ma et al., 2012; Kosuri & Church, 2014; Shagin et al., 2017). Amplification through PCR further reshapes error distributions, with systematic biases associated with GC content, fragment length, and cycle number (Aird et al., 2011; Benjamini & Speed, 2012). Sequencing technologies also contribute heterogeneous error modes: Illumina platforms are dominated by substitutions, Nanopore

sequencing produces indel clusters, and duplex sequencing reduces random noise while distinguishing true mutations (Schmitt et al., 2012).

Motivated by natural language processing, recent work has modeled DNA as a language. Transformer-based pretraining such as DNABERT (Ji et al., 2021; Zhou et al., 2024) and the Nucleotide Transformer (Dalla-Torre et al., 2025) demonstrates broad transfer across genomics tasks. Convolutional models including DeepSEA (Zhou & Troyanskaya, 2015), Basset (Kelley et al., 2016), and BP-Net (Avsec et al., 2021) highlight inductive biases for motif discovery, while state-space models emphasize efficient handling of long-range dependencies (Zhu et al., 2024). Although primarily developed for functional genomics tasks such as accessibility, transcription factor binding, and expression prediction, these backbones have not been systematically evaluated for *technical error*.

Beyond end-to-end sequence models, another line of work augments deep embeddings with hand-crafted descriptors such as GC bias, homopolymer length, or positional context (Zhou & Troyanskaya, 2015; Kelley et al., 2016). While informative, naïve concatenation often introduces redundancy, unstable optimization, and scale mismatches, which limit predictive stability and interpretability. Feature attribution studies (e.g., SHAP, integrated gradients) highlight the value of importance-guided selection for reducing noise, yet systematic strategies for integrating descriptors and embeddings remain underexplored in genomics. Similar challenges arise in multimodal and tabular learning, where principled methods for combining structured descriptors with learned embeddings are still an open problem. Moreover, neural predictors in biology, as in other domains, are frequently overconfident. Post-hoc calibration techniques such as temperature scaling are simple and effective (Guo et al., 2017), with extensions to regression and predictive intervals (Kuleshov et al., 2018).

Miscalibration has also been widely documented in computer vision and natural language processing (Ovadia et al., 2019), motivating research on reliable uncertainty estimation. However, calibration quality typically degrades under distribution shift. Despite increasing awareness of dataset bias and perturbation sensitivity, few genomics models explicitly address robustness across experimental conditions such as aging, PCR protocols, or sequencing platform changes. These limitations motivate the development of approaches that unify feature refinement, principled fusion, and calibration to improve reliable per-base error prediction in synthetic biology.

## 3 PROPOSED METHOD

To overcome these issues, we design **REFORM**, a lightweight framework that refines descriptors, integrates them with embeddings through stable fusion, and adapts predictive confidence to distribution shifts. Although applicable to settings where structured descriptors accompany learned embeddings, we instantiate the framework for *DNA storage pipelines*, where descriptors such as GC content, homopolymer length, and structural accessibility provide a natural and challenging testbed. The following subsections describe the components and design choices of REFORM in detail.

### 3.1 INPUT REPRESENTATION AND FEATURE REFINEMENT

In addition to backbone DNA language-model embeddings, we explicitly incorporate handcrafted descriptors that capture biochemical and structural risk factors. We construct an over-complete pool of 51 features grouped into four categories:

1. *Global composition and length.* GC ratio, sequence length, dinucleotide bias, and molecular weight approximate overall chemical stability. High GC ratios, for example, are empirically linked to PCR bias and secondary structure formation.

2. *Motif statistics.* Frequencies of di-/tri-nucleotides, homopolymer length, and start/stop-like codons. Long homopolymers are known to cause polymerase slippage, producing indels in Nanopore sequencing.

3. *Structural proxies.* Sliding-window GC variance, Shannon entropy, repeat counts, and palindromic subsequence length. These features approximate local folding tendencies and predict sequencing dropout in regions with high secondary structure.

4. *Positional and accessibility cues.* Relative position within the oligo, flanking sequence composition, one-hot base identity, and predicted accessibility from RNAplfold. Errors are often enriched near termini or structurally constrained regions.

We deliberately construct redundancy in this pool, since discarding potentially causal proxies a priori risks losing signals critical for error prediction. To obtain a compact and stable representation, we refine the 51-dimensional feature block using two complementary, metric-aligned importance estimators.

*(i) SHAP-based attribution.* We train a lightweight logistic probe on the training split and compute SHAP values $\phi_{i,j}$ for feature $j$ on validation instance $i$. Feature importance is defined as the mean absolute SHAP value:

$$a_j \;=\; \frac{1}{|\mathcal{D}_{\mathrm{val}}|} \sum_{i \in \mathcal{D}_{\mathrm{val}}} |\phi_{i,j}|,$$

yielding a global ranking that is model-agnostic yet faithful to predictive contributions.

*(ii) PR–AUC drop analysis.* We evaluate feature relevance with respect to the discrimination metric. Let $M(\cdot)$ denote PR–AUC; we compute the baseline score $M_{\mathrm{base}}$ and measure the expected drop when feature $j$ is permuted:

$$d_j = M_{\mathrm{base}} - \mathbb{E}_\pi[M(\pi_j(\mathbf{f}))],$$

where $\pi_j$ permutes feature $j$ across samples, repeated $K$ times to reduce variance. This ensures alignment between feature ranking and the evaluation objective.

*(iii) Redundancy-penalized selection.* We aggregate the two rankings into a normalized score and select a subset of fixed size $k$ by solving

$$S^\star \;\in\; \arg \min_{S \subset [m],\, |S|=k} \left( \sum_{j \in S} s_j + \beta \sum_{\substack{j < k \\ j,k \in S}} |C_{jk}| \right),$$

where $C$ is the empirical correlation matrix and $\beta > 0$ penalizes redundancy. In practice we set $k = 9$, which balances predictive coverage and computational cost.

The refined vector is $\boldsymbol{f}_{\mathrm{ref}} = (\hat{f}_j)_{j \in S^\star}$, with per-feature min–max normalization to ensure comparability. The refinement procedure is run on training folds only, and the selected indices $S^\star$ are reused for validation and test to prevent leakage. In our experiments $S^\star$ has size $k{=}9$ and remains fixed prior to training. Appendix A.3 provides mechanistic justifications linking each retained feature to biochemical error processes (e.g., GC-driven folding, repeat-induced slippage). To provide intuition, Figure 2 visualizes feature rankings from PR–AUC drop analysis. These results illustrate complementary perspectives on feature relevance; quantitative comparisons and ablations are presented in Sec.4.

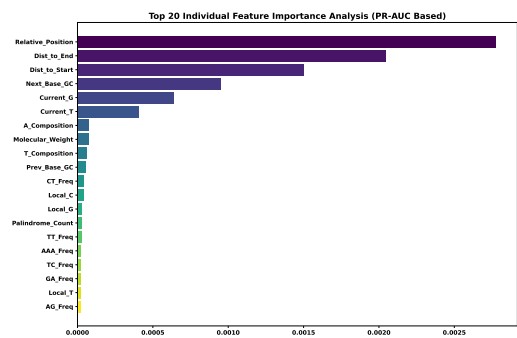

Figure 2: Feature importance analysis.

## 3.2 Cross-attentive Fusion with Contextual Embeddings

Backbone embeddings capture long-range sequence context, while handcrafted descriptors encode localized biochemical risk. Direct concatenation of these channels is unstable: heterogeneous scales can cause feature dominance and redundancy. To address this, we design a *cross-attentive, convex vector-gated fusion* block. Specifically, let $\boldsymbol{h}_{\mathrm{cls}} \in \mathbb{R}^d$ denote the backbone [CLS] token. The refined descriptor vector $\boldsymbol{f}_{\mathrm{ref}}$ is projected into a token set $U \in \mathbb{R}^{m \times d}$. We form a query $\mathbf{q} = W_q \boldsymbol{h}_{\mathrm{cls}}$, keys $\mathbf{K} = U W_k$, and values $\mathbf{V} = U W_v$, and compute a cross-attention readout:

$$\boldsymbol{r} = \mathrm{softmax}\left(\frac{\mathbf{q}\mathbf{K}^\top}{\sqrt{d_k}}\right) \mathbf{V}.$$

A gating vector is then defined as

$$\boldsymbol{g} = \sigma\big(G[\boldsymbol{h}_{\mathrm{cls}}; \boldsymbol{r}]\big) \in (0,1)^d,$$

and the fused representation is

$$\boldsymbol{z} = \boldsymbol{g} \odot \boldsymbol{h}_{\mathrm{cls}} + (1 - \boldsymbol{g}) \odot \boldsymbol{r}.$$

**Theorem 1.** *Assume $\|\boldsymbol{h}_{cls}\| \leq M$, $\|U\| \leq B$, and all linear maps have operator norm $\leq L$. With a row-stochastic softmax, the readout satisfies*

$$\|\boldsymbol{r}\| \leq BL, \quad \|\boldsymbol{z}\| \leq M + BL.$$

*Moreover, the mapping $(\boldsymbol{h}_{cls}, U) \mapsto \boldsymbol{z}$ is Lipschitz as a composition of bounded linear maps, softmax (locally Lipschitz), sigmoid, concatenation, and elementwise affine operations. Hence the fused representation is globally bounded and stable.*

**Corollary 1.** *As a direct consequence of the gating formulation, for each coordinate $\ell$, $z_\ell \in \mathrm{conv}\{h_\ell, r_\ell\}$ since $g_\ell \in (0, 1)$. This ensures that every dimension is an interpolation between contextual and descriptor channels, eliminating coordinate-wise explosion and complementing the global boundedness guarantee.*

The block therefore interpolates between contextual and descriptor channels, ensuring stability absent in concatenation or FiLM-style modulation. Cross-attention allows descriptors to selectively attend to contextual features, while the gating vector adaptively balances their contributions. In practice, $\boldsymbol{f}_{\text{ref}}$ is projected by a two-layer MLP with ReLU activation into $U \in \mathbb{R}^{m \times d}$. We employ 8-head cross-attention, each head operating on $d/8$ dimensions, followed by concatenation and linear projection back to $\mathbb{R}^d$. The gate is instantiated as $\boldsymbol{g} = \sigma(W[h_{\text{cls}}; r] + b)$ with dropout ($p = 0.1$) for regularization and LayerNorm applied after fusion. This design introduces $O(md + d^2)$ additional parameters relative to the backbone.

Given the fused embedding $\boldsymbol{z}$, REFORM predicts the error probability as

$$p = \sigma(\boldsymbol{w}_c^\top \boldsymbol{z}).$$

Since error labels are highly imbalanced, with positives much rarer than negatives, we use the focal loss to emphasize informative positives:

$$\mathcal{L}_{\text{focal}} = -\alpha y(1 - p)^\gamma \log p - (1 - \alpha)(1 - y)p^\gamma \log(1 - p).$$

To promote stability across conditions, we add a supervised NT-Xent loss. Unlike unsupervised contrastive learning, positive pairs are defined as samples with the same label ($y_i = y_j$), while negatives are those with different labels. For a minibatch $\mathcal{B}$, the loss is

$$\mathcal{L}_{\text{sup-NT-Xent}} = \frac{1}{|\mathcal{B}|} \sum_{i \in \mathcal{B}} -\log \frac{\sum_{j \neq i} \mathbb{1}[y_i = y_j] \exp(\cos(\boldsymbol{z}_i, \boldsymbol{z}_j)/\tau)}{\sum_{k \neq i} \exp(\cos(\boldsymbol{z}_i, \boldsymbol{z}_k)/\tau)},$$

with $\ell_2$-normalized embeddings and cosine similarity. The final objective then combines the two terms:

$$\mathcal{L} = \mathcal{L}_{\text{focal}} + \lambda \mathcal{L}_{\text{sup-NT-Xent}}.$$

Here, the focal term addresses label imbalance, while the contrastive term aligns representations of same-label bases across conditions and repels mismatched pairs. In practice, we apply minibatch-level undersampling to reduce skew, with focal weighting providing further correction. Standard cross-entropy loss is recovered as a special case ($\gamma = 0, \lambda = 0$). Empirically, both focal and contrastive terms are necessary for stable training under severe imbalance and condition shifts (see Sec.4).

## 3.3 CONDITION-AWARE CALIBRATION

Neural predictors are often overconfident, and their calibration quality deteriorates under distribution shifts. In DNA storage pipelines, experimental conditions such as PCR cycles or storage duration introduce systematic but modest distributional changes. To address this, we perform *per-condition calibration* using temperature scaling.

Given fused logit $s = \boldsymbol{w}_c^\top \boldsymbol{z}$ and condition $c$, the calibrated probability is

$$p' = \sigma\left(\frac{s}{\tau_c}\right), \qquad \tau_c > 0,$$

where $\tau_c$ is a condition-specific temperature. Each $\tau_c$ is estimated on a held-out calibration split by minimizing negative log-likelihood or Brier score. This preserves ROC–AUC invariance under monotone transforms while improving probability calibration.

Table 1: Performance comparisons under different configurations.

| Configuration | ROC–AUC | PR–AUC | F1 | ECE |
|---|---|---|---|---|
| DNABERT [CLS] | 0.7799 | 0.8703 | 0.7873 | 0.1930 |
| + Accessibility | 0.7802 | 0.8706 | 0.7583 | 0.1615 |
| + 36+14 (E1: window) | 0.7821 | 0.8712 | 0.7698 | 0.1587 |
| + 36+14 (E2: multi-scale) | 0.7852 | 0.8729 | 0.7764 | 0.1424 |
| + 36+14 (E3: context-conditioned) | 0.7870 | 0.8741 | 0.7781 | 0.1398 |
| Refined 9-D (selected) | 0.7891 | 0.8756 | 0.7802 | 0.1365 |
| + attention fusion (full FT) | 0.7920 | 0.8795 | 0.7792 | 0.1467 |

Other post-hoc calibrators include Platt scaling and isotonic regression. However, both approaches are prone to overfitting in low-sample regimes, which is common in DNA experiments. Temperature scaling is parameter-efficient and stable with as few as hundreds of examples, making it a natural choice for condition-aware calibration.

To further stabilize estimates, we adopt a hierarchical prior that ties each $\tau_c$ to a global baseline $\tau_0$:

$$\tau_c^{\star} \in \arg\min_{\tau>0} \ \mathbb{E}_{(s,y)\sim\mathcal{D}_{\text{cal}}^{(c)}}\big[\mathcal{S}(\sigma(s/\tau), y)\big] + \lambda_{\text{reg}}(\tau - \tau_0)^2.$$

At inference time, predictions are routed by condition metadata (e.g., PCR cycle count, storage duration) and adjusted with the corresponding $\tau_c$. Unseen conditions default to $\tau_0$ or can be quickly adapted with a few calibration samples.

## 4 EXPERIMENTS

### 4.1 EXPERIMENTAL SETUP

We evaluate REFORM on the task of per-base error prediction in DNA storage pipelines. Rather than introducing a new backbone, REFORM augments existing sequence models such as DNABERT and the Nucleotide Transformer with lightweight modules for feature refinement, fusion, and calibration. Experiments are conducted on a representative DNA storage dataset with systematically varied laboratory conditions. The digital twin dataset provides Illumina SBS reads from 40 sequencing experiments with different synthesis, amplification, and aging protocols (Gimpel et al., 2023). To complement these data, we also generate simulated reads using ART (Huang et al., 2012), matching error models and coverage to the digital twin dataset. This allows controlled *in silico* perturbations that extend robustness evaluation beyond dataset-specific artifacts.

Different synthesis, amplification, and aging protocols naturally induce diverse error profiles: aging introduces hydrolytic and oxidative lesions, while high-cycle PCR amplifies GC bias and polymerase slippage. To prevent leakage, data are split by sequence identity, and we adopt a leave-one-condition-out (LOCO) protocol, where models trained on nominal conditions are evaluated on held-out stress-test conditions. This setup enables systematic assessment of distribution shift and highlights the advantages of feature refinement, vector-gated fusion, and condition-aware calibration. We report discrimination metrics (PR–AUC, ROC–AUC, F1) and calibration metrics (ECE with $M = 15$ bins, Brier score). PR–AUC is the primary metric due to severe class imbalance, and all results are averaged over three random seeds.

### 4.2 EXPERIMENTAL RESULTS

**Step 1: From DNABERT to Refined Features and Attention Fusion.** We begin with a DNABERT [CLS] classifier as a baseline and augment it with explicit biological descriptors. The initial feature block includes 51 descriptors spanning three categories: (i) a scalar *accessibility* feature, (ii) 36 sequence-derived descriptors (e.g., GC content, k-mer frequencies, repeat statistics, entropy measures), and (iii) 14 position-derived descriptors (e.g., relative coordinates, local base composition, distances to sequence termini). We explore three extraction strategies: (E1) center-aligned sliding windows, (E2) multi-scale windows $\{11, 31, 51\}$ aggregated by mean and max, and (E3) context-conditioned projections guided by hidden states.

Table 3: Performance of REFORM combined with different backbones under LOCO evaluation.

| Backbone | Params (M) | PR–AUC (in) | PR–AUC (LOCO) | ECE (in) | ECE (LOCO) |
|---|---|---|---|---|---|
| DNABERT | 91.97 | 0.8102 | 0.8109 | 0.0762 | 0.0855 |
| Nucleotide Transformer (50M) | 57.23 | 0.9132 | 0.9147 | 0.1223 | 0.1239 |
| Nucleotide Transformer (100M) | 99.22 | 0.9189 | 0.9184 | 0.0952 | 0.0960 |
| Nucleotide Transformer (250M) | 237.89 | 0.9278 | 0.9291 | 0.0944 | 0.0968 |

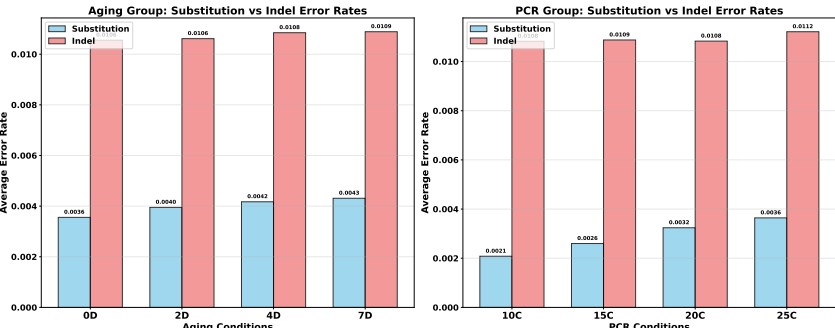

Figure 4: Empirical error distributions under aging (left) and PCR (right).

Attribution and SHAP analysis revealed strong redundancy among the 50+ features. A consistent subset of **nine** descriptors was identified, comprising five GC/complexity features (GC_Content, Sequence_Entropy, Local_GC_Mean, Local_GC_Std, Max_Repeat_Length), three positional/structural features (Dist_to_Start, Dist_to_End, Palindrome_Count), and one global feature (Sequence_Length). This compact representation improved PR–AUC and calibration stability, showing that targeted refinement is both effective and interpretable. Figures 2 confirm that refined descriptors achieve attribution scores comparable to, or exceeding, individual Transformer units, reflecting known biochemical error mechanisms. We then investigate two integration regimes: (i) *refine-only*, which concatenates the 9-D vector with contextual embeddings, and (ii) *attention-fusion*, where token-wise gates dynamically balance Transformer embeddings and refined features. As shown in Table 1, explicit feature integration consistently improves DNABERT, while refinement enhances stability. The *Refined 9-D* setting achieves the strongest calibration (ECE 0.1365), and attention fusion yields the best discrimination (ROC–AUC 0.7920, PR–AUC 0.8795), with minor trade-offs in F1 and ECE. We therefore adopt the *Refined 9-D + attention fusion (full FT)* configuration as the backbone.

**Step 2: Loss Function Analysis.** We first assess how different training objectives affect discrimination and calibration under the DNABERT + Accessibility setting. Standard cross-entropy (CE), focal loss, and focal loss with a supervised NT-

Table 2: Loss analysis under DNABERT + Accessibility.

| Loss Function | F1 | PR–AUC | ROC–AUC |
|---|---|---|---|
| CE loss | 0.784 | 0.842 | 0.771 |
| Focal loss | 0.786 | 0.870 | 0.780 |
| Focal + supervised NT-Xent | 0.814 | **0.882** | **0.793** |

Xent term are compared to isolate the effects of reweighting and representation alignment. As shown in Table 2, focal loss substantially improves PR–AUC and ROC–AUC relative to CE, confirming the benefit of addressing imbalance. Adding the contrastive term further enhances discrimination, though with a slight drop in F1 due to stronger emphasis on hard negatives.

We also analyze sensitivity to hyperparameters. Following prior work (Lin et al., 2017; Wang & Liu, 2021), we use $(\alpha = 0.25, \gamma = 2)$ for focal loss and $\tau = 0.5$ for NT-Xent, with `pos_weight=2.0`, `neg_weight=1.0`, and `contrastive_loss_weight=0.1`. And Figure 3 shows validation curves: PR–AUC increases steadily to a peak of **0.8782** by epoch 15, while F1 converges near **0.7894** by epoch 9 before stabilizing. These results indicate that focal terms mitigate imbalance, moderate $\tau$ stabilizes alignment, and a small but non-zero $\lambda$ improves robustness without overwhelming the focal objective. We adopt these settings in subsequent experiments and leave broader hyperparameter optimization for future work.

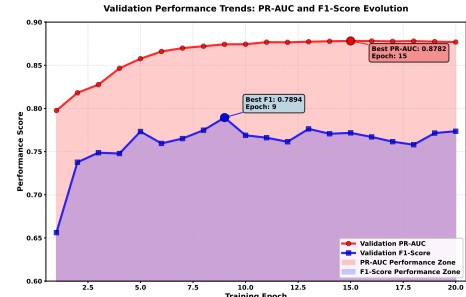

Figure 3: Validation performance trends under representative hyperparameter settings.

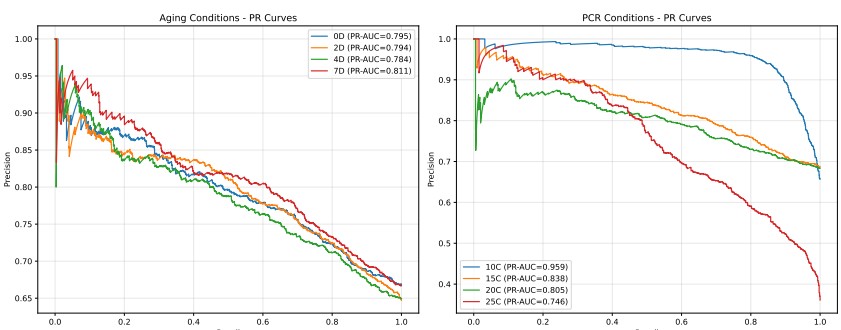

Figure 5: Per-condition PR curves under aging (left) and PCR (right).

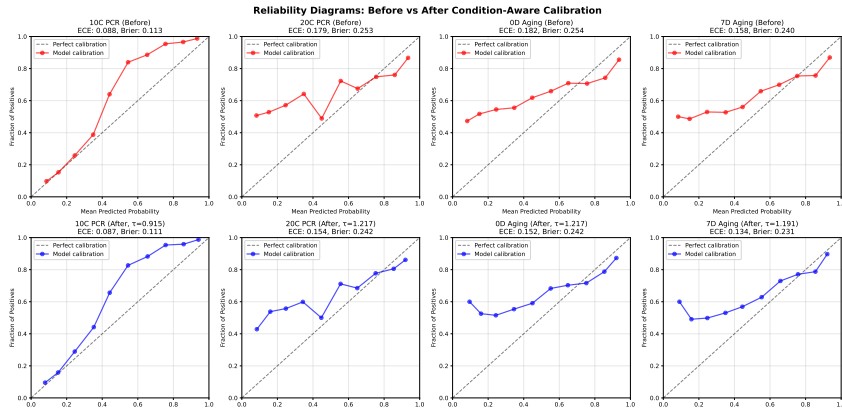

Figure 6: Reliability diagrams before and after condition-aware calibration.

**Step 3: Comparison across Backbone Scales.** Because REFORM is lightweight and modular, it can be coupled with sequence backbones of different sizes without altering their pretraining. We evaluate DNABERT and three Nucleotide Transformer variants (50M, 100M, and 250M parameters) under the same LOCO protocol, with all backbones augmented by REFORM through refined features and convex vector-gated fusion. As shown in Table 3, REFORM provides consistent gains across models of varying capacity. Larger backbones achieve stronger absolute scores, yet even the smallest model (Nucleotide Transformer, 50M) benefits significantly from our modules. These findings confirm that REFORM does not replace large-scale pretraining but acts as a transferable risk prediction layer that complements DNA language models and improves calibration under distribution shifts.

**Step 4: Stress-Test Inference and Calibration.** We evaluate inference under aging and PCR stress tests. Aging spans 0, 2, 4, and 7 days, while PCR spans 10, 15, 20, and 25 cycles. These regimes introduce systematic perturbations to error patterns, although the overall distributions remain close to the nominal setting. Figure 4 illustrates the observed shifts.

Figure 5 shows the per-condition PR curves. For aging, PR–AUC remains stable across 0D, 2D, and 4D (∼0.79), but increases at 7D (0.811). This suggests that prolonged degradation strengthens error signatures and improves separability. For PCR, the trend is reversed. Performance is highest at 10 cycles (PR–AUC=0.959), but decreases with further amplification (15C=0.838, 20C=0.805, 25C=0.746). This indicates that heavy amplification introduces complex and heterogeneous error modes, such as GC bias and polymerase slippage, which reduce precision especially at high recall. Taken together, aging improves separability, while PCR amplifies uncertainty, underscoring the importance of condition-aware calibration in downstream analyses.

Figure 6 compares reliability diagrams before and after applying condition-aware calibration. Across all regimes, calibration reduces both ECE and Brier score, confirming that post-processing improves the reliability of predicted probabilities. For PCR, the benefit is limited at low cycle counts (10C: ECE 0.088→0.087), since the model is already well calibrated, but becomes more pronounced at higher cycles (20C: ECE 0.179→0.154), where amplification-induced errors distort probability estimates. For aging, improvements are more consistent: 0D improves from ECE 0.182 to 0.152 and 7D from 0.158 to 0.134, with Brier scores decreasing in parallel. These results indicate that calibration is

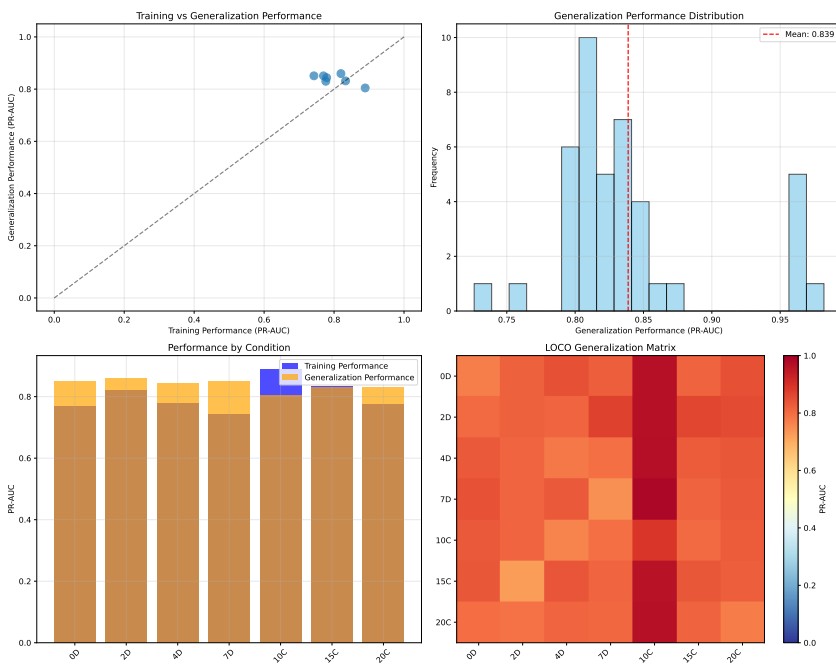

Figure 7: LOCO generalization analysis.

especially effective when error modes are structured and predictable, while under high-cycle PCR the residual miscalibration reflects the greater heterogeneity of error patterns.

Figure 7 summarizes cross-condition generalization under the LOCO protocol. Average LOCO PR–AUC remains high (∼0.84), confirming that the model retains strong discriminative power when transferred to unseen regimes. Aging-induced error patterns generalize well across time points, reflecting their more regular degradation processes. In contrast, PCR amplification is more challenging: while 10-cycle PCR transfers reliably, higher-cycle settings exhibit clear drops in PR–AUC, consistent with the increased variability caused by GC bias and polymerase slippage. The LOCO matrix also shows that generalization is relatively stable within condition families (aging↔aging, PCR↔PCR), but deteriorates more sharply across families, especially from aging to high-cycle PCR.

## 5  CONCLUSIONS AND LIMITATIONS

In conclusion, this work introduces REFORM, a lightweight framework for per-base error prediction under distribution shifts in DNA storage pipelines. The framework integrates refined feature selection, cross-attentive gated fusion, and condition-aware calibration. Systematic evaluation on digital twin and simulated datasets shows that REFORM consistently increases PR–AUC and reduces calibration error across different backbones and experimental settings. Stress-test analyses further indicate distinct behaviors under aging and PCR conditions: aging amplifies error patterns and improves separability, while PCR introduces heterogeneous biases that reduce calibration quality. Together, these findings suggest that explicit integration of refined descriptors with contextual embeddings provides a principled way to achieve trustworthy base-level predictions. However, several limitations remain. The refined features are selected using importance analyses but still rely on predefined biological proxies, and incorporating richer physical or chemical models may yield further gains. The calibration strategy depends on explicit condition metadata such as PCR cycle count or storage duration, yet in practice such information may be incomplete or noisy. The evaluation is restricted to DNA storage pipelines, and broader applications in synthetic biology and functional genomics require further validation. Future work should explore adaptive feature discovery that does not rely on predefined descriptors, calibration methods that remain robust with partial or uncertain metadata, and extensions to new domains to confirm the generality of the framework.

## REPRODUCIBILITY STATEMENT

We have made several efforts to ensure the reproducibility of our results. The datasets used in this work are publicly available: the Digital Twin dataset (Gimpel et al., 2023), with preprocessing steps described in Section 4. Simulation recipes were generated by strictly following the experimental conditions defined in the Digital Twin dataset. Implementation details of **REFORM**, including model architectures, training objectives, and calibration procedures, are provided in Section 3.

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

## A  APPENDIX

### A.1  REPRODUCIBILITY

If this paper is accepted, we will release the full implementation of **REFORM** on GitHub under an open-source license. During the rebuttal period, if reviewers request access to our code, we will provide it through an anonymized GitHub repository to facilitate a fair and thorough evaluation.

### A.2  THE USE OF LARGE LANGUAGE MODELS (LLMS)

We made limited use of large language models during manuscript preparation. Their role was restricted to language polishing and minor style adjustments. All conceptual ideas, algorithmic designs, theoretical results, and experimental analyses presented in this work are entirely original and developed by the authors.

### A.3  FEATURE IMPORTANCE, SELECTION RATIONALE, AND MECHANISTIC THEORY

This appendix formalizes why the retained non-contextual features are mechanistically effective for per-base error prediction. We connect each feature group to stylized error mechanisms under simple physical or probabilistic models. Let $p_i$ denote the per-base error probability at position $i$, with $p_i = \sigma(s_i)$ where $s_i = \boldsymbol{w}^\top \boldsymbol{z}_i$ depends on local sequence context and the retained descriptors. We now describe monotone or convex relationships between $p_i$ and each feature group.

**GC Content, Local Mean/Variance and Entropy.** Let $g_i \in [0, 1]$ be the local GC fraction in a window around $i$. Secondary-structure occupancy $\pi_i$ is often modeled as $\pi_i \propto \exp(-\Delta G(g_i)/RT)$, where $\Delta G(g_i)$ combines stacking and hydrogen-bond contributions. Using a linearized energy $\Delta G(g) = a - bg$ with $b > 0$, we obtain

$$\pi_i \ \propto \ \exp(-a/RT) \exp(bg_i/RT),$$

which is increasing and convex in $g_i$. If the error hazard rises with structure occupancy, then $p_i$ increases convexly with $g_i$. By Jensen's inequality, higher variance of $g_i$ increases expected error at fixed mean. Local Shannon entropy $H_i$ also contributes. Under homopolymer-driven slippage, error risk is higher for skewed composition (low entropy). A simple model is

$$p_i = \alpha\, \phi(g_i) + \beta\, \psi(H_i) + \cdots, \qquad \psi'(H) \le 0,$$

so $p_i$ decreases with entropy. Hence *GC content*, *Local_GC_Mean*, *Local_GC_Std*, and *Sequence_Entropy* form a coherent set of descriptors.

**Theorem 2** (GC convexity and variance sensitivity)**.** *Let $p_i = h(g_i)$ where $h$ is increasing and convex. Then, for any window,*

$$\mathbb{E}[p_i] \ \ge \ h(\mathbb{E}[g_i]), \qquad \frac{\partial \mathbb{E}[p_i]}{\partial \mathrm{Var}(g_i)} \ \ge \ 0.$$

*Thus both* Local_GC_Mean *and* Local_GC_Std *are informative: the mean reflects the convex baseline, while variance amplifies risk under heterogeneous GC content.*

*Proof.* By Jensen's inequality, $\mathbb{E}[p_i] \ge h(\mathbb{E}[g_i])$. Writing $g_i = \bar{g} + \epsilon$ with $\mathbb{E}[\epsilon] = 0$ and expanding $h(\bar{g} + \epsilon)$ shows $\mathbb{E}[p_i]$ increases with $\mathrm{Var}(\epsilon)$ if $h'' \ge 0$. $\qquad\square$

**Max Repeat Length and Slippage.** Let $r_i$ be the maximal homopolymer run length at $i$. Under a discrete-time slippage model with per-step probability $q \in (0, 1)$, the probability of at least one event is

$$P_{\text{slip}}(r) = 1 - (1-q)^r,$$

which is increasing and concave in $r$. Thus *Max_Repeat_Length* is a monotone driver of risk.

**Lemma 1** (Monotonicity in run length). *For $r_1 < r_2$, $P_{slip}(r_1) < P_{slip}(r_2)$, and*

$$\frac{\partial P_{slip}}{\partial r} = (1-q)^r \log \frac{1}{1-q} > 0.$$

**Palindromes and Hairpins.** Let $K_i$ be the number of palindromic substrings (length $\geq \ell_0$) overlapping $i$. For palindrome $k$, hairpin probability is $\pi_k \propto \exp(-\Delta G_k / RT)$. Assuming independence,

$$P(\text{hairpin near } i) \geq 1 - \prod_{k=1}^{K_i}(1 - \pi_k),$$

which increases with $K_i$. Hence *Palindrome_Count* correlates with structure-induced errors.

**Boundary Effects: Distance to Termini.** Let $x \in [0, L]$ be the base coordinate and define $d_{\text{start}} = x/L$, $d_{\text{end}} = (L - x)/L$. A boundary-layer hazard model,

$$\lambda(x) = \lambda_0 + \lambda_b\big(e^{-x/\xi} + e^{-(L-x)/\xi}\big), \quad \lambda_b, \xi > 0,$$

implies higher error near both ends. Thus *Dist_to_Start* and *Dist_to_End* act as monotone predictors.

**Theorem 3** (Monotone end effects). *Under the boundary-layer hazard model, per-base risk decreases with both $d_{start}$ and $d_{end}$, achieving its minimum near the sequence center.*

**Global Length.** If motifs such as palindromes or homopolymers occur as a Poisson process with rate $\rho$, then for a sequence of length $L$ the probability of at least one problematic motif in a neighborhood around $i$ is bounded by

$$P(\exists \text{ motif near } i) \leq 1 - \exp(-c\rho L),$$

with $c > 0$ capturing interaction range. Hence *Sequence_Length* captures global exposure to structural or synthesis-cycle risks not explained by local descriptors.

## A.4 REFORM TRAINING PIPELINE

As illustrated in Algorithm 1, the training pipeline of REFORM follows four main stages. First, we refine handcrafted features from the initial 51-dimensional pool by combining SHAP-based importance scores, permutation drop analyses, and redundancy penalties, resulting in a compact 9-dimensional descriptor vector. Second, we integrate refined features with backbone embeddings through cross-attentive convex vector-gated fusion, which adaptively balances contextual and descriptor information. Third, we train the fused predictor using a composite objective that combines focal loss to address label imbalance with a supervised NT-Xent loss to align representations of samples with the same label across conditions. Finally, we calibrate predicted probabilities per condition by learning condition-specific temperatures with a hierarchical prior, using a held-out calibration split. At inference, each sequence is processed through feature refinement and fusion, followed by condition-aware calibration to produce reliable per-base error probabilities.

**Algorithm 1** REFORM: Feature-Refined Fusion and Condition-Aware Calibration

**Require:** Training corpus of DNA sequences with per-base labels $\{(x^{(n)}, y^{(n)})\}_{n=1}^{N}$; condition labels $c^{(n)} \in \mathcal{C}$; backbone encoder $\text{Backbone}(\cdot)$; initial handcrafted feature extractor $\text{Feat51}(\cdot) \in \mathbb{R}^{51}$; target subset size $k{=}9$; redundancy weight $\beta > 0$; loss weights $(\alpha, \gamma, \lambda)$; contrastive temperature $\tau$.

**Ensure:** Calibrated predictor $x \mapsto p'(x \mid c)$ with per-condition temperatures $\{\tau_c\}_{c \in \mathcal{C}}$.

1: **Split data** into train/val/test by sequence (and by condition when needed). Reserve a small *calibration split* $\mathcal{D}_{\text{cal}}^{(c)}$ per condition $c$ from the *training side only*.                     ▷ // prevent leakage

2: **function** FEATUREREFINEMENT($\mathcal{D}_{\text{train}}, \mathcal{D}_{\text{val}}$)
3:        For each sample, compute raw features $\mathbf{f} \in \mathbb{R}^{51}$ with $\text{Feat51}(\cdot)$.
4:        Train a lightweight logistic probe on $\mathcal{D}_{\text{train}}$ using $\mathbf{f}$.
5:        On $\mathcal{D}_{\text{val}}$, compute SHAP values $\phi_{i,j}$ and set $a_j \leftarrow \frac{1}{|\mathcal{D}_{\text{val}}|} \sum_i |\phi_{i,j}|$.
6:        Compute baseline PR–AUC $M_{\text{base}}$ on $\mathcal{D}_{\text{val}}$; for each feature $j$, estimate $d_j \leftarrow M_{\text{base}} - \mathbb{E}_\pi \big[ M\big(\pi_j(\mathbf{f})\big) \big]$ by permuting coord. $j$ across samples (repeat $K$ times).
7:        Rank each feature by $a_j$ and $d_j$ (descending); aggregate normalized ranks to get $s_j$.
8:        Estimate correlation matrix $C$ on *training* features; select
$$S^\star \in \arg\min_{S \subset [51],\, |S|=k} \left( \sum_{j \in S} s_j + \beta \sum_{j<k,\, j,k \in S} |C_{jk}| \right).$$
9:        Fit per-feature min–max scalers on *training* for $j \in S^\star$; define $\mathbf{f}_{\text{ref}} \in \mathbb{R}^k$ by applying the same affine maps to val/test and clamping to $[0, 1]$.
10:        **return** $S^\star$, scalers, and a function $\text{Refine}(x) = \mathbf{f}_{\text{ref}}$.
11: **end function**

12: **function** FUSION($x$)
13:        $\mathbf{h}_{\text{cls}} \leftarrow \text{Backbone}(x)$                                          ▷ // contextual embedding in $\mathbb{R}^d$
14:        $\mathbf{f}_{\text{ref}} \leftarrow \text{Refine}(x)$; project to tokens $U \in \mathbb{R}^{m \times d}$ (linear + optional positional codes).
15:        $\mathbf{q} \leftarrow W_q \mathbf{h}_{\text{cls}}, \; \mathbf{K} \leftarrow U W_k, \; \mathbf{V} \leftarrow U W_v$
16:        $\mathbf{r} \leftarrow \text{softmax}\big(\frac{\mathbf{q}\mathbf{K}^\top}{\sqrt{d_k}}\big)\mathbf{V}$                              ▷ // cross-attn readout
17:        $\mathbf{g} \leftarrow \sigma\big(G[\mathbf{h}_{\text{cls}}; \mathbf{r}]\big) \in (0, 1)^d$
18:        **return** $\mathbf{z} \leftarrow \mathbf{g} \odot \mathbf{h}_{\text{cls}} + (1 - \mathbf{g}) \odot \mathbf{r}$
19: **end function**

20: **function** TRAIN($\mathcal{D}_{\text{train}}$)
21:        **for** epoch $= 1, \ldots, E$ **do**
22:            **for** minibatch $\mathcal{B} \subset \mathcal{D}_{\text{train}}$ (apply random under-sampling of majority class) **do**
23:                For each $(x_i, y_i, c_i) \in \mathcal{B}$: $\mathbf{z}_i \leftarrow \text{Fusion}(x_i)$; $s_i \leftarrow \mathbf{w}_c^\top \mathbf{z}_i$; $p_i \leftarrow \sigma(s_i)$.
24:                **Focal loss:** $\mathcal{L}_{\text{focal}} \leftarrow \sum_i \big[ -\alpha y_i (1-p_i)^\gamma \log p_i - (1-\alpha)(1-y_i) p_i^\gamma \log(1-p_i) \big]$.
25:                **Supervised NT-Xent:** define positives $\{(i,j) : y_i{=}y_j, i{\neq}j\}$; use cosine on $\ell_2$-normalized $\mathbf{z}$:
$$\mathcal{L}_{\text{sup}} \leftarrow \frac{1}{|\mathcal{B}|} \sum_{i \in \mathcal{B}} -\log \frac{\sum_{j \neq i,\, y_j = y_i} \exp(\cos(\mathbf{z}_i, \mathbf{z}_j)/\tau)}{\sum_{k \neq i} \exp(\cos(\mathbf{z}_i, \mathbf{z}_k)/\tau)}.$$
26:                **Total:** $\mathcal{L} \leftarrow \mathcal{L}_{\text{focal}} + \lambda \mathcal{L}_{\text{sup}}$; update parameters by Adam/SGD.
27:            **end for**
28:        **end for**
29: **end function**

30: **function** CALIBRATE($\{(s, y, c)\}_{\text{from } \mathcal{D}_{\text{cal}}}$)
31:        **for** each condition $c \in \mathcal{C}$ **do**
32:            Solve $\tau_c^\star \in \arg\min_{\tau > 0} \mathbb{E}_{(s,y) \sim \mathcal{D}_{\text{cal}}^{(c)}} \big[ \mathcal{S}(\sigma(s/\tau), y) \big] + \lambda_{\text{reg}} (\tau - \tau_0)^2$
            where $\mathcal{S}$ is NLL or Brier; $\tau_0$ is a global prior (shared) temperature.
33:        **end for**
34:        **return** $\{\tau_c^\star\}_{c \in \mathcal{C}}$
35: **end function**

36: **function** INFER($x$, condition $c$)
37:        $\mathbf{z} \leftarrow \text{Fusion}(x)$; $s \leftarrow \mathbf{w}_c^\top \mathbf{z}$; **if** $c$ unseen **then** use $\tau_0$ **else** use $\tau_c^\star$.
38:        **return** $p'(x \mid c) \leftarrow \sigma(s/\tau_c^\star)$
39: **end function**
40: **Run:** $(S^\star, \text{Refine}) \leftarrow \text{FEATUREREFINEMENT}(\mathcal{D}_{\text{train}}, \mathcal{D}_{\text{val}})$; $\text{TRAIN}(\mathcal{D}_{\text{train}})$; $\{\tau_c^\star\} \leftarrow \text{CALIBRATE}(\mathcal{D}_{\text{cal}})$.

