# OpenReview forum: "ReFoRM: Reliable Per-Base Error Prediction under Distribution Shifts in DNA Storage"
_ICLR.cc/2026/Conference — ICLR 2026 Conference Withdrawn Submission_

### Official Review · Reviewer_rejD · 2025-10-19

**Soundness:** 3
**Presentation:** 2
**Contribution:** 2
**Rating:** 2
**Confidence:** 3

**Summary:**

This work proposes a method to refine an initial pool of 51 handcrafted features into a 9-dimensional subset. These features are selected for their potential influence on the error rates that arise during DNA synthesis, PCR, and sequencing. For any given DNA sequence, this refined feature vector is combined with contextual embeddings from a pre-trained language model, such as DNABERT, using a cross-attentive, vector-gated fusion module. The resulting fused representation is then used to predict the per-base error probability along the sequence.

**Strengths:**

+ This work is well-designed and methodologically sound.
+ Comprehensive and experimental evaluation.

**Weaknesses:**

While the proposed framework is a valuable contribution to the specific task of per-base error prediction, its novelty for the broader machine learning community appears incremental. The work's primary contribution seems to be the thoughtful application and integration of established techniques to a new domain, rather than the development of fundamentally new methods.

Specifically, the module for selecting important descriptors leverages a combination of conventional methods. Furthermore, the proposed gated mechanism for fusing handcrafted features with embeddings from DNABert is based on architectural principles that are well-established in the deep learning field.

**Questions:**

+ For clarity and accessibility, acronyms such as SHAP, PR-AUC, and LOCO should be defined upon their first use in the manuscript.
+ The diagram in Figure 1 could be misleading, as it does not visually connect the input DNA sequence to the feature extraction process for the handcrafted features. This makes the refined vector, $f_{ref}$, appear to be independent of the input.
+ The equations between lines 166-187 contain several typographical inconsistencies that should be revised. For example, the font for the feature vector $f$ is inconsistent.
+ The necessity and practical implication of Theorem 1 and Corollary 1 are not sufficiently justified. They seems have little relation with the proposed work.
+ There appears to be a minor typesetting issue on Line 247, where the indicator function symbol ($\mathbb{1}$) has not been rendered correctly, likely due to a missing or conflicting LaTeX package.
+ The performance improvements reported in Table 1 across different module configurations are marginal. This raises questions about the practical necessity and cost-benefit of adding several complex modules for such incremental gains.
+ The DNABert and Nucleotide Transformer backbones were pre-trained on biological DNA sequences, which have different statistical properties from the random or encoded sequences used in DNA storage. How this domain shift between pre-training and application affect the overall performance?
+ The procedure for defining a ground truth error label ($y_i=1$) is not detailed. Given that error occurrences in DNA pipelines can be stochastic and fall within a 1% to 10% range, relying purely on observation may introduce substantial label noise. The methodology for establishing ground truth should be clarified.
+ How sequence alignment and per-base labeling are handled following an insertion or deletion event. This is a crucial detail, as indels disrupt the one-to-one correspondence between the designed and sequenced DNA.
+ More details should be presented regarding the "lightweight logistic probe" used for feature selection.

---

### Official Review · Reviewer_6NZ6 · 2025-10-28

**Soundness:** 3
**Presentation:** 3
**Contribution:** 3
**Rating:** 6
**Confidence:** 3

**Summary:**

This paper introduces REFORM, a lightweight framework for reliable per-base error prediction in DNA storage. REFORM is designed for solving distribution shifts from varying experimental conditions. The framework first refines a compact set of informative features from a large pool, then uses a stable cross-attention mechanism to fuse these features with embeddings from a sequence model like DNABERT, and finally applies condition-aware calibration to improve prediction reliability. Experiments demonstrate that REFORM significantly enhances both accuracy (PR-AUC) and calibration (ECE) when generalizing to unseen conditions, proving its robustness against distribution shifts.

**Strengths:**

- Clear Motivation: The paper clearly reveals the challenge of per-base error prediction under distribution shifts in DNA storage and proposes a corresponding framework to address it. This is meaningful to the community.
- Comprehensive Evaluation: The authors conduct a comprehensive evaluation with the LOCO protocol. Stress tests and evaluation of discrimination (PR-AUC) and calibration (ECE, Brier score) confirm robust empirical validation.
- Feature Refinement: The framework employed a dual-metric strategy which combines SHAP, PR-AUC drop and redundancy penalty. This combination distills a highly informative and compact 9-feature subset.

**Weaknesses:**

- Component Originality: In this paper, the originality of the individual components is limited because feature selection, attention, and temperature scaling are all established techniques. The primary contribution is the thoughtful integration of these components, which seems incremental rather than innovative.
- Limited Generality: The paper claims that the proposed framework is broadly applicable but it implicitly restricts itself to encoder-based architectures. Specifically, the main pipeline in this work requires an encoder model to support prediction. However, recent advances in genomic foundation models focus primarily on generative tasks. State-of-the-art models such as Evo2[1] and GenomeOcean[2] are designed with autoregressive, decoder-only fashion. Given the rapid adoption of decoder-only architectures in genomic foundation models, it would be valuable for the authors to discuss how REFORM could be adapted to work with such models?
- Backbone Selection: In the experimental setup, the authors use DNABERT and three versions of the Nucleotide Transformer (NT) as backbone models. However, even for encoder-only classification genomic foundation models (GFMs), there are more advanced models like DNABERT2 and NT2, which incorporate architectural and design improvements. These newer versions have consistently shown better performance on classification benchmarks. Could the authors clarify why the older backbone models were chosen, and whether the framework’s performance would differ when using these stronger GFMs?

[1] Genome modeling and design across all domains of life with Evo 2.

[2] GenomeOcean: An Efficient Genome Foundation Model Trained on Large-Scale Metagenomic Assemblies.

**Questions:**

See the weaknesses section.

---

### Official Review · Reviewer_RUhd · 2025-10-30

**Soundness:** 2
**Presentation:** 2
**Contribution:** 2
**Rating:** 2
**Confidence:** 3

**Summary:**

This paper presents ReFoRM, a framework to predict base-level error probabilities in DNA data storage pipelines, even under distribution shifts (e.g. different PCR cycles or aging conditions).
It refines a large initial set of 51 sequence descriptors down to 9 key features (e.g. GC content, repeats, distances to ends) using SHAP attributions and performance-drop analysis.
These selected features are then fused with a DNABERT sequence embedding via a cross-attention module and a convex gating mechanism.
This ensures each output is a bounded convex combination of the deep [CLS] embedding and the feature-based representation.
Finally, condition-aware calibration is applied: each experimental condition gets a learned temperature scaling to adjust the model’s output probabilities for better calibration under shift.

Experiments on a digital twin DNA storage dataset (with varied synthesis, amplification, and storage conditions) show that ReFoRM achieves high discrimination (PR–AUC ~ 0.92) and low calibration error (ECE ~ 0.09) on both in-distribution and leave-one-condition-out tests. It outperforms a DNABERT baseline and ablations without feature refinement or multi-domain calibration.

**Strengths:**

- Effective integration of domain features with deep sequence context.

- High predictive performance under distribution shifts (PR–AUC ~0.93, with minimal drop out-of-domain).

- Improved probability calibration (ECE ~0.09) via simple, parameter-efficient scaling.

**Weaknesses:**

- [major] Limited novelty: The method mostly combines existing ideas (SHAP-based feature selection, cross-attentive gating, and temperature scaling) rather than introducing fundamentally new algorithms or theory.

- [major] Baselines and ablations: No comparison to a simpler descriptor-only model (e.g. a logistic or XGBoost on the 9 features) is provided, so it’s unclear how much the DNABERT context adds beyond known sequence features. Likewise, the paper omits baselines like deep ensembles or domain-adaptation methods that could handle distribution shift.

- [major] Dependence on metadata: ReFoRM requires the condition label at inference to route inputs to the correct temperature scaling. This reliance on known domains means an unseen condition must default to a global temperature, which may not fully correct miscalibration. The approach lacks an unsupervised way to detect or adapt to entirely novel shifts.

- [minor] Calibration baseline unclear: The authors claim Platt scaling and isotonic regression overfit in low-data regimes, but they do not quantitatively show these alternatives. It’s hard to gauge how much condition-specific scaling outperforms a single global calibration, since no direct comparison or reliability diagrams are given.

- [minor] Statistical reporting: Results are averaged over 3 runs, but no error bars or significance tests are reported. This makes it difficult to assess variance or confirm that improvements (e.g. ECE reductions) are statistically reliable.

- [minor] Reproducibility and code: Code and data are not provided with the submission (though promised in the appendix). While authors provides methodology and implementation details, there still are some guesswork needed to reproduce the results. Hence, I'd like to examine the code and hence request the code.

- [major] Presentation: there are some presentation issues. for example, many figures completely unreadable. font size is too small. Also, eqn `line247` has clear error (my guess is indicator function?). Overall language can also be better as well.

- [major] theory: I am having problem to see the necessity and actual relevance of the theoretical results in this paper. They are mostly vacuous, trivial and decoupled from the core method. They offers no guarantees for ECE, PR–AUC, or shift robustness. I suggest the authors to remove them, or, if the authors deem their necessity, please add discussions to stress their relevance precisely.

**Questions:**

please see above

LLM disclaimer: I use LLM to polish my language and understand some cited refs.

---

### Official Review · Reviewer_oCwY · 2025-10-30

**Soundness:** 3
**Presentation:** 2
**Contribution:** 1
**Rating:** 4
**Confidence:** 2

**Summary:**

This paper presents REFORM, a modular framework for reliable per-base error prediction in DNA storage pipelines. The method consists of three components: (1) feature refinement combining SHAP-based attribution and PR–AUC importance; (2) a cross-attentive convex vector-gated fusion block and (3) condition-aware calibration using per-condition temperature scaling. Experiments report improved calibration and PR-AUC under various distribution shifts.

**Strengths:**

1. The structure of the framework (feature → fusion → calibration) is clear and modular.
2. The empirical results show improvements on meaningful metrics.
3. The writing and figures are generally clear.

**Weaknesses:**

1. While the paper is well engineered, almost all components are adaptations of existing methods: Feature refinement relies on SHAP and permutation-based feature importance, Fusion builds on standard cross-attention and gating/FiLM-like modulation, Calibration is a direct extension of temperature scaling  with per-condition parameters and a regularization term. Therefore, REFORM does not introduce a fundamentally new algorithmic idea, but rather a task-specific integration of existing building blocks.
2. Although the Introduction briefly cites prior work on attention, calibration, and feature descriptors, the core method section (section 3) provides no in-text references when reusing these techniques. For instance, the SHAP-based importance computation, the permutation-drop analysis, and temperature scaling are described as if they were proposed here. This can mislead readers and the real novelty is not clear.
3. The motivation for combining these particular components is mostly empirical; the paper does not clearly argue why this combination yields new insights beyond engineering convenience.

**Questions:**

1. Could the authure clarify what is genuinely novel beyond combining existing methods?
2. Why are the original sources (Lundberg & Lee 2017; Guo et al. 2017; Vaswani et al. 2017; Perez et al. 2018) not cited in Section 3 where those components are introduced?

---

### Author Response · Authors · 2025-11-12

We would like to sincerely thank the reviewers, area chair, and senior area chair for their time and valuable feedback on our submission. We have carefully read and reflected on all the comments. Although we have decided to withdraw this paper from ICLR 2026 due to the current evaluation scores, we truly appreciate the constructive insights provided throughout the review process. These suggestions will help us refine and extend our work in future revisions.

Thank you again for your effort and professionalism in reviewing our paper.

---

### Note · Authors · 2025-11-12

I have read and agree with the venue's withdrawal policy on behalf of myself and my co-authors.